# Headwaters to valley: Water quality in rivers transitioning from forest to agricultural bottomland

Doug Graber Neufeld*, Isaac Alderfer, Zachary Bauman, Micah Buckwalter

Department of Biology, Eastern Mennonite University, Harrisonburg, Virginia, United States of America

* neufeldd@emu.edu

## Abstract

Many waterways flow out of forestlands, which tend to maintain higher water quality, into agricultural lands, which tend to degrade water quality. The roles of land cover in impacting key water quality parameters (phosphorus, nitrogen, total suspended solids, bacteria, and conductivity) were investigated for the watershed of the North and South Fork of the Shenandoah River, Virginia. This area has a particularly sharp boundary between heavily forested and heavily agricultural regions. Two datasets were analyzed: 1) a large number of datapoints spanning a 20-year range in the Water Quality Portal (WQP) database, and 2) transects along three representative rivers systems over the span of a 4-year period. All parameters trended better in forested regions than agricultural regions. This was particularly true for nitrogen and conductivity; phosphorus, TSS and bacteria showed more local variability, especially in the agricultural region. Periods of high flow increased phosphorus, sediment and bacteria concentrations, and decreased conductivity, but not when drainage basin forest cover was less than 80%. Transects showed that waterways flowing out of forestland maintained higher water quality for approximately the first 8 km in agricultural land. Both transect and WQP data indicated higher water quality when the percent of forested land cover in a drainage basin was about 70–80%. Thus, forestland does mitigate the impacts of agriculture on water quality to some degree, but this effect rapidly diminishes as forest cover of the watershed lessens. Furthermore, forests themselves have degraded water quality at certain times and places; for instance, nutrients level were in the medium to high stress level for aquatic life in approximately 15% of samples. This study illustrates general trends of land cover effects on water quality, while also highlighting both site-specific variability, and the dynamics of water quality as water flows out of forested areas into agricultural areas.

**Data availability statement:** All water quality data files are available from the zenodo database (DOI 10.5281/zenodo.14218942); https://doi.org/10.5281/zenodo.15865614.

**Funding:** This work was funding as research awards from Eastern Mennonite University (emu.edu) to DGN. The funders had no role in study design, data collection and analysis, decision to publish, or preparation of the manuscript.

**Competing interests:** The authors have declared that no competing interests exist.

## Introduction

Aquatic ecosystems from headwaters to larger waterways downstream have been a major focus of efforts at maintaining and improving environmental quality, due to their combined importance in providing ecosystem services, and their integral role in the lives of people [1]. Indeed, maintenance of water quality is directly related to important planetary boundaries such as the geochemical cycling of nitrogen and phosphorus [2], which have clearly exceeded safe operating zones. Numerous studies have explored key water quality parameters elucidating both patterns that are generalizable across waterways, and patterns that are specific to local spatial and temporal conditions.

Nutrients (nitrogen and phosphorus), sediments, and bacteria are recognized as key parameters determining both local aquatic health, and downstream impacts [3,4]. While water quality studies have elucidated key insights on the dynamics of these pollutants in watersheds, these studies also point towards a continued need to understand these dynamics better. This need is highlighted by observations that concentrations of these pollutants generally remain high, despite long-running efforts to reduce the loading of these water contaminants through management actions. For instance, a recent study looked at thousands of water sample measurements found that the high nutrients were due to the presence of persistent sources, including the potential role of legacy sources or inadequate current management practices [5]. That study highlighted the role of local ecological conditions as potentially key to controlling watershed level pollution, and confirmed the important role of headwaters in determining downstream concentrations. Studies using large datasets have noted the sustained increase at the continental scale, such as the study of Stoddard et al. [6] finding the percentage of stream length with <10ug/L total phosphorus (TP) decreased from 24.5 to 1.6% in the 2004–2014 period. Likewise, more focused use of local datasets note a similar lack of progress at the level of local watersheds, such as the study on three Chesapeake Bay watersheds finding a lack of improvement in nutrient and sediment levels despite the implementation of multiple management practices [7]. The slow progress in improving water quality highlights the need to for continued understanding of sources, transport, fate and impacts of these pollutants. In particular, understanding the key drivers of these processes, and the accompanying management practices to mitigate water quality impacts, is an ongoing need.

Land cover, and especially continued pressure from human activities that alter land cover, are fundamental in driving water quality of watersheds. Most notable is how natural vegetation such as forests can improve water quality, while agricultural activities (both pastureland and cropland) can degrade water quality through a variety of processes. In the Chesapeake Bay watershed, agriculture is the primary suspected cause of degraded water quality [8], whereas forest cover can improve water quality [9]. Protection of forested lands is generally seen as crucial to maintaining clean water [10,11], notably illustrated by a recent study in the southeast United States finding that forests had the highest water quality of all landscape types [12]. In parallel to land cover effects, headwater regions (which often are more forested)

have unique characteristics compared to downstream waters [13–15]. There are clearly multiple factors beyond just land cover type that control water quality [12], and more information is needed on how water quality changes as waterways traverse different landscapes. In particular, it is not clear whether there is a sharp transition in water quality as it traverses forested and agricultural land, and how the impacts of the two landscapes combine to influence downstream water quality.

Watersheds draining into the Chesapeake Bay are the focus of a long running effort to understand how stressors such as nutrients and sediments contribute to estuarine health, and how these inputs can be reduced [16–18]. Efforts in the Chesapeake watershed have been mixed, illustrating both the ongoing challenge of reducing agricultural inputs, and the need for more information on what exactly drives nutrient trends [16,18,19]. Land cover modification has been highlighted as one of the key drivers in the Chesapeake Bay watershed [9]. The Shenandoah River watershed of Virginia and West Virginia is a major tributary system of the Chesapeake Bay, and stands out as having both some of the region's largest protected forestlands, and also the highest density of livestock and poultry operations in Virginia, along with significant cropland. Furthermore, there is a generally distinct boundary between the relatively flat valley bottomland which is dominated by agriculture and predominantly of limestone origin, compared to the surrounding mountains which are heavily forested, federally protected as national forest and national park land, and predominantly of sandstone origin. This watershed therefore represents a unique opportunity to study a sharp transition of waterways between heavy forestland of headwaters, and intensive agriculture downstream.

The present study is unique in combining data from a large public database (Water Quality Portal; WQP) with targeted sampling on representative river systems to understand the dynamics of key water contaminants in a headwaters watershed of the Chesapeake Bay with distinct regions of land cover. We used this system to focus on several interrelated questions that add novel insights to ongoing efforts at understanding nutrient, sediment and bacteria dynamics: 1) What characterizes water quality in heavily forested headwater regions? 2) How does this water quality change when transitioning from forested to agriculturally intensive lands? and 3) What are similarities and differences between different water quality parameters in this pattern of landscape?

## Methods

### Ethics statement

Site access outside of national forest was either at publicly accessible right-of-way along state or county highways (usually bridges), or by landowner permission. Within the national forest, permission for site access for the purpose of water sampling was granted by the US Forest Service.

### North and South Fork Shenandoah River watershed

Water quality data for the combined watersheds of the North and South Forks of the Shenandoah River watershed were accessed from the Water Quality Portal (WQP; https://www.waterqualitydata.us/), an open access compilation of water quality results from multiple government, academic and nonprofit organizations. Additional publicly available data was added from the Friends of the Shenandoah River (https://fosr.org/state-of-the-river/compiled-water-quality-data/), which runs a Virginia state approved laboratory and has collected significant water quality data for the region. Since land downstream of the confluence of the North and South Forks (north of Front Royal, Virginia) does not have protected National Forest or National Park land (as dominates forestland upstream from this point), we did not analyze the watershed of the Shenandoah River downstream of the confluence. Samples from within the delineated watershed were selected for the 20 year period from Jan 1, 2004 to Dec 31, 2023. Since most sites had multiple measurements of a parameter for that period, we used the mean parameter value for each site.

In order to separate the effects of flow (*e.g.,* base flow vs storm), we analyzed data from USGS water stations (USGS National Water Dashboard; https://dashboard.waterdata.usgs.gov/app/nwd/en/) at the base of the North Fork (#01634000) and South Fork (#01635500). Although flow at water sample points on subwatersheds depends on the

local conditions, these gauges roughly correlated with flow at gauges upstream in both watersheds (upstream North Fork #01632000 correlated with an $r^2$ of 0.55, and upstream South Fork #01620500 correlated with an $r^2$ of 0.42), suggesting they are representative of the watershed. Thus, in the absence of flow data at all of the many water sample points, the downstream flow rates provided an estimate of when there were high flow periods throughout the watershed. Water sample measurements were selected as "high flow" when the corresponding large watershed (North or South Fork) had a discharge in the 75th percentile or above for the corresponding USGS water gauge (percentiles reported by USGS for a 99 year period). Most of these times (80% and 77% in North and South Forks, respectively) occurred during the December to May period. The remaining samples (below the 75th percentile) were designated "low flow".

Outliers of parameters were identified and removed for samples having Cook's distance of 4/n (0.005 for this data-set), and $p < 0.05$. The cause of outlying data was not known, it could have been inaccurate measurements or genuinely unusual high or low levels that only occurred occasionally. Given the varied sources of data in the Water Quality Portal and large number of samples, removal of these outliers ensured that occasional quality control issues (inaccurate measurements or mis-recorded data), or unusually large fluctuations of parameters did not greatly skew measurements representing the majority of water flow periods for a waterway.

Spatial analysis was performed using ArcGIS Pro 3.3 and QGIS 3.38.0. Since in other aquatic systems, water quality is shown to be driven by overall land use across the entire drainage basin for a collection site [20], we delineated watersheds from USGS 1/3 arc-second Digital Elevation Maps (3DEP product in The National Map). The digital elevation maps were processed in QGIS to fill and remove sinks, followed by flow direction and accumulation calculations. The unnest watersheds function (Whitebox) in QGIS was used to delineate watersheds representing the entire drainage basin for each water sample point. Land use/land cover (LULC) from the USGS National Land Cover Database was compared between 2003 and 2021, and showed a low degree of change in forested vs unforested land (in either direction) during this time (0.96%). We therefore used the 2021 land cover data set (Fig 1a) for calculating percent forest in all watersheds, rather than attempting to match year of water collection with the respective year of LULC data. Geology was mapped as Generalized Lithography in the State Geologic Map Compilation (SGMC) Geodatabase of the Conterminous United States (ver. 1.1, August 2017), and was categorized as karst or non-karst geology (Fig 1b). Soil Erodibility was mapped as the K factor from the USA soil service geographic database (SSURGO) of the Natural Resources Conservation Service (accessed as ArcGIS Online layer) (Fig 1c). The percent forestland and karst, and the average erodibility, in a drainage basin represented by sample sites reflects both site location (for instance, whether it is in a forested vs nonforested area), and the area upstream draining into the point. Water quality values were compared to watershed forest coverage, karst coverage, and average soil erodibility (K factor) using multiple linear regression. Further comparisons of parameters in discrete forest coverage categories (in 20% increments) were performed with a Kruskal Wallis and Dunn's post-hoc test (non-parameter statistics were used due to unequal variances in categories).

To calculate a general forest boundary in the study area, we calculated the average percent forest within a 200m radius of each pixel (pixel resolution 25m) of the land cover raster using the focal statistics tool of ArcGIS Pro. Pixels with at least 75% forest in this radius were counted as forest. This value was chosen as it produced a smoothed general forest boundary that depicts the visual forest transition as seen on satellite images. River kilometers for the transect study (see below) were then calculated upstream and downstream from the point at which the river crosses this boundary.

Cluster and outlier identification was performed in ArcGIS using Anselin Local Moran's I using 999 permutations. Anselin Local Moran's I is a spatial statistical method used to identify areas of interest in other studies of water quality [21,22]. Statistically significant clusters and outliers at the 95 percent confidence level are shown as clusters of high values (HH), clusters of low values (LL), outliers where a high value is mostly surrounded by low values (HL), and outliers where low value is mostly surrounded by high values (LH).

## A. Land Cover

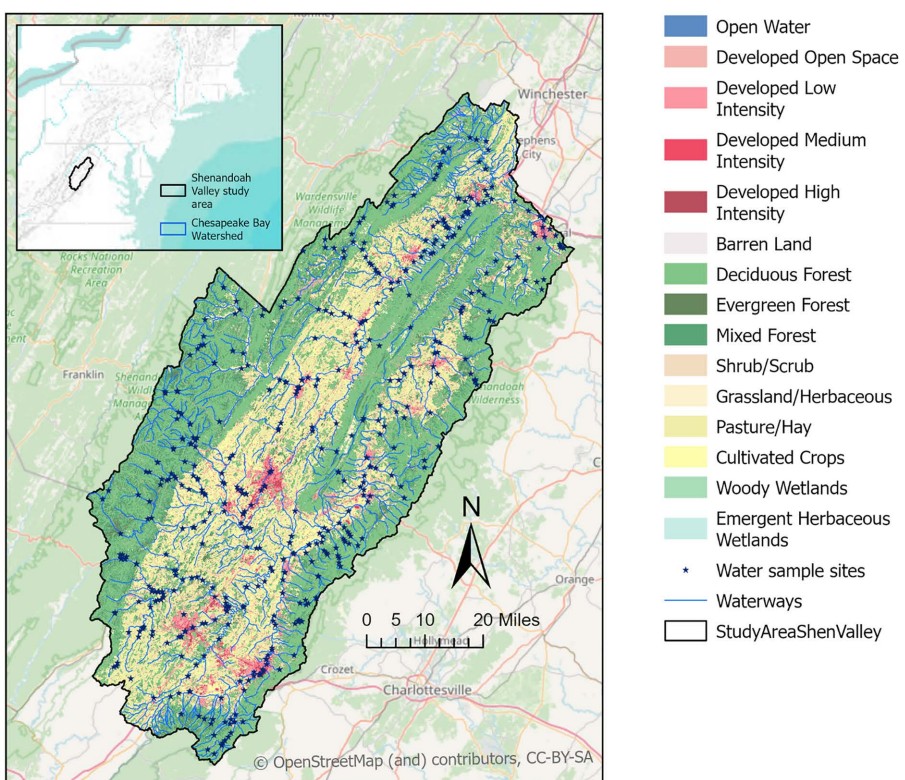

## B. Erodibility

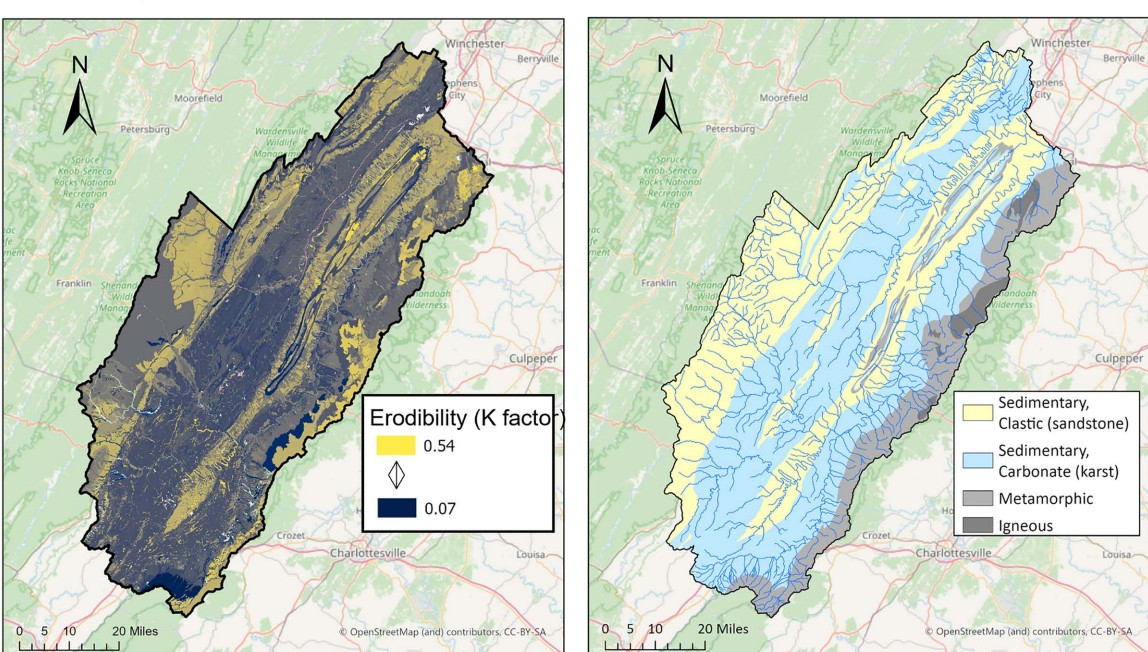

**Fig 1. Land cover (A), erodibility (B) and underlying lithology (C) in study area.** These three parameters represent candidate drivers of water quality, and highlight the difference between the valley area (central, running southwest – northeast), and the mountain areas (on the western and eastern edges, and in the central Massanutten ridge region). Study area represents watersheds of the North and South Forks of the Shenandoah River. Base map and data from OpenStreetMap and OpenStreetMap Foundation.

## River transect sites

Three river systems were chosen for water sampling as transects – the Dry River, Briery Branch, and the North River (Fig 2). All three rivers originate in the mountains on the western edge of the Shenandoah watershed, converge as the lower North River, and then flow together into the South Fork of the Shenandoah River which joins the Potomac River before entering the Chesapeake Bay. The headwaters are at approximately 3000 feet, and flow through the oak-hickory forest of George Washington National Forest before abruptly entering the Shenandoah valley agricultural region. The land cover characteristics are generally similar between the three river systems, with the overall watersheds dominated by forest and agriculture, with only a small fraction of each watershed as developed land or wetlands (Fig 2).

Water sample sites were selected at approximately 5 km intervals, although specific sites were dependent on availability of river access (see Fig 2). In addition, river sections in the middle interior of the valley dried completely during periods of low rainfall, especially in midsummer. In other cases of high rainfall or winter, sites in the national forest were inaccessible for sampling. The final number of samples per site therefore varied. Samples were collected between March 2020 and July 2024 and were taken at all times of the year (see S1 Table for specific dates). As a general indication of river level in the region, flow discharge was recorded from a USGS station (#01622000) near Burketown on the North River, and varied

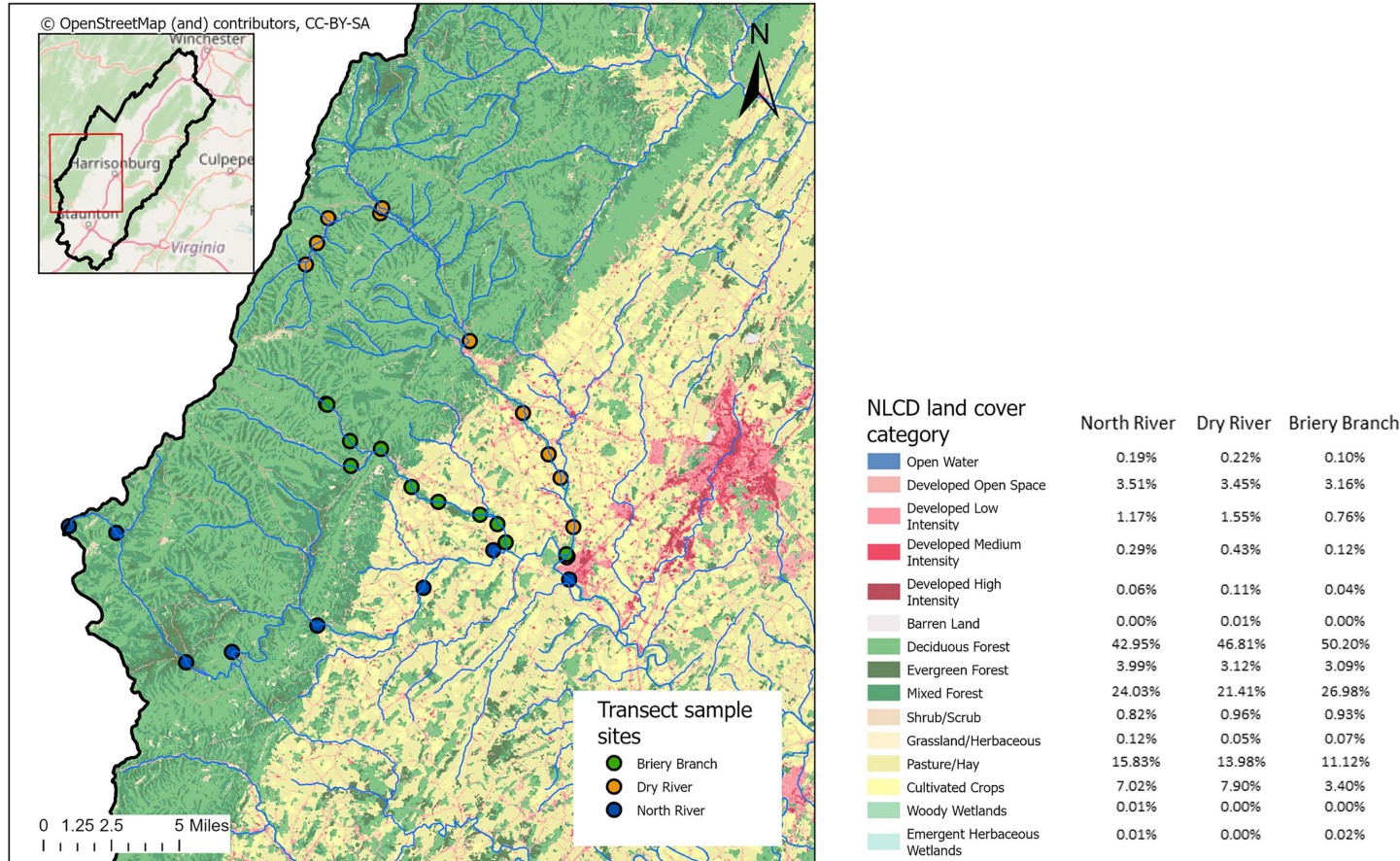

**Fig 2. Location and characteristics of river transect.** Map of river transect sites, located on the western side of the overall watershed study area, along with land cover categories for the region. Percentages of land cover categories in each river watershed are shown in the legend, indicating the overall similarity of the three watersheds used in the transect study. Base map and data from OpenStreetMap and OpenStreetMap Foundation.

from 114 to 1430 ft$^3$ per second, representing a range of discharge from 18th to 97th percentiles for the period samples were taken. Samples thus represent a wide range of seasons and river flow conditions.

At each site, parameters were either measured onsite (temperature and conductivity), or water samples were collected in dedicated water sample bottles (nitrate, orthophosphate, bacteria, sediments). All measurements or bottle samples were collected in the main column below the water surface, upstream from any disturbances (wading or previous sampling). Field blanks were collected at one site per transect for nitrate, orthophosphate, and bacteria. A field spike bottle was also collected at one site per transect for nitrate and orthophosphate. Field duplicates were collected for all nitrate, orthophosphate and bacteria samples, and for one sediment sample per transect. All samples were kept on ice (for transport) or at 4°C storage until analysis.

### River transect parameters

Water samples were analyzed for sediment, nitrate, orthophosphate, temperature, conductivity, and general and fecal coliforms. A Quality Assurance Project Plan, previously approved for earlier NFWF and Virginia DEQ grant sampling, guided analysis, and was based on USGS, Virginia DEQ, or EPA approved procedures (e.g., U.S. EPA 2007. SW-846).

Sediments were measured gravimetrically as per EPA method 160.2. In brief, 1 L samples were filtered through pre-rinsed, dried and weighted 47 mm diameter GF/F filters (0.7 μM pore-size), and then re-dried and re-weighed. Conductivity was measured using a Tracer Pocketester (LaMotte) calibrated to a 84 μS/cm standard. Probe calibration was verified after each transect to ensure calibration did not drift during the measurement period. Temperature was measured using a Vernier temperature probe, or the internal thermometer on the conductivity probe. General and fecal coliforms were measured using Coliform Easygel (Micrology, Inc; Goshen, Indiana, USA). 1 ml samples were plated on agar petri dishes, incubated at 37°C until colony color development (generally 36-48h), and then colony identity determined by color.

Nitrogen and phosphorus were assessed as nitrate and orthophosphate, respectively. While other forms would be present, these are the most common and relevant forms in these waters. For instance, nitrate is the majority of total nitrogen (TN) in the region [4]. Likewise, the study of Farthing et al [23] on the effect of forestland on agriculturally impacted rivers found most nitrogen as nitrate, and most phosphorus as phosphate. Both nitrate and orthophosphate were measured with a colorimetric assay (Hach; for NitraVer5 and PhosVer3, respectively) according to manufacturer instructions. Duplicate sample bottles for each site were measured and the mean used as the final value. Absorbance values at appropriate wavelengths were compared to a calibration series, with blank and standard tested every 10th sample.

## Results

### Water quality patterns in the North and South Forks of Shenandoah River watershed

Water quality data for the entire watershed of the North and South Forks of the Shenandoah River were compared with several candidate drivers of water quality – documented land cover changes (USGS National Land Cover Database) due to human activities (represented by percent forest coverage), underlying geology (represented by karst coverage, as a rock type in the area having a large impact on hydrology), and basic soil composition (represented by erodibility, as a measure of how likely soil constituents are to enter waterways) (Fig 1). Multiple linear regression identified forest coverage as more important for driving water quality, compared to karst or soil erodibility (Table 1) for all parameters except conductivity, which highly correlated with both forest coverage and soil erodibility, and moderately correlated with karst. The relatively low adjusted r$^2$ values point towards a complex system with a large for additional factors contributing to water quality. None-the-less, the relationship of parameters to forest coverage was highly statistically significant, whereas it was not for karst or soil erodibility (aside from conductivity). The predominance of land cover effects compared with measured geology and soil differences confirms other studies noting the importance of land cover, and further analysis focused on forest coverage as a major driver of water quality.

**Table 1. Multiple Linear Regression of water quality parameters compared with land cover (forest coverage), geology (karst coverage), and soil (average erodibility) in corresponding watersheds.**

| Water Quality Parameter | sample size (N) | adjusted r² | Forest coverage | | Karst | | Erodibility | |
|---|---|---|---|---|---|---|---|---|
| | | | significance (p value) | coefficient | significance (p value) | coefficient | significance (p value) | coefficient |
| Nitrate | 297 | 0.59 | **p<0.001** | −2.8 | p=0.12 | −0.41 | p=0.65 | −0.003 |
| TN | 230 | 0.54 | **p<0.001** | −3.34 | p=0.06 | −0.67 | p=0.11 | −0.026 |
| Orthophosphate | 211 | 0.07 | **p<0.01** | −0.06 | p=0.32 | −0.02 | p=0.91 | −0.0001 |
| TP | 266 | 0.07 | **p<0.01** | −0.07 | p=0.18 | −0.07 | p=0.30 | −0.001 |
| TSS | 209 | 0.22 | **p<0.001** | −16.8 | p=0.21 | −4.83 | p=0.40 | −0.15 |
| *E. coli* | 265 | 0.18 | **p<0.01** | −769 | p=0.87 | 42.8 | p=0.07 | −24.6 |
| Conductivity | 636 | 0.75 | **p<0.001** | −563 | p=0.08 | 46.1 | **p<0.001** | 2.71 |

Mapped concentration averages at the sample sites highlights several general patterns (Fig 3). First, concentrations of all parameters were consistently lower in forested regions, although there was variability with some sites having moderate concentrations. Second, the highest concentrations of all parameters were clearly centered in the valley, although the pattern differed between parameters. For some (nitrate, TN, and conductivity), the concentrations in the valley were consistently high, whereas for the other parameters they were more variable in the valley. Third, there was a gradient with the highest values most often located more centrally in the valley, rather than on the forest edges. In addition, some low concentrations along the valley edge (the boundary with forests) were associated with waterways entering the valley from the forest. Conductivity showed the most consistent gradient, showing a gradual increase with distance from the forest into the valley. Finally, different parameters often showed similar trends at sites (for instance, a site with a high concentration of one parameter often had high concentrations of other parameters) suggesting generalized water quality issues at specific sites or areas of the valley.

Cluster and outlier analysis (Anselin Local Moran's I) was performed to statistically investigate these spatial distributions. This confirmed that clusters of low concentrations tended to be located in heavily forested land, but in some cases were just outside the forestland, or along major waterways that would take significant volumes of water into the valley (Fig 4; light blue points). Clusters of high values were located in the valley, usually at some distance from the forested areas (Fig 4; light red points). Cluster analysis also confirmed the more consistent pattern of nitrogen (nitrate, and TN) and conductivity relative to land cover, whereas other parameters showed more variability (for instance, with clusters of low concentrations located in the valley). For all parameters, Anselin Local Moran's I identified some outliers (high values in low cluster areas, dark red points; or low values in high cluster areas, dark blue points), pointing to significant localized variation. In particular, there were low outlier sites (Fig 4; dark blue points) scattered in the valley for different parameters, indicating some local areas of higher water quality among a region with generally low water quality. Outlier high values (Fig 4; dark red points) were less common, suggesting there were not as many point sources that only impacted a small area – problem areas tended to be across a stretch of waterway or region (such as the central area of the valley).

Given that parameters may relate nonlinearly to the proportion of forest cover, we further analyzed the relationship by grouping watersheds into 5 bins representing 20% increments of forest cover. Furthermore, since storm events are typically major drivers for loads of the water contaminants, we analyzed data separately for high flow (discharge volumes at or above the 75th percentile), and for low flow (discharge volumes below the 75th percentile). Statistical analysis (Kruskal Wallis test, followed by post-hoc Dunn's test) indicated significant differences with forest cover for all parameters (Fig 5) at both high and low flows. The greatest statistical difference was between the highest category of forest cover (80–100%) and the other forest cover categories. This was especially evident, for instance, with Nitrate, TP and TSS, which generally had statistically similar concentrations between the forest cover categories, except for the highest category (80–100%)

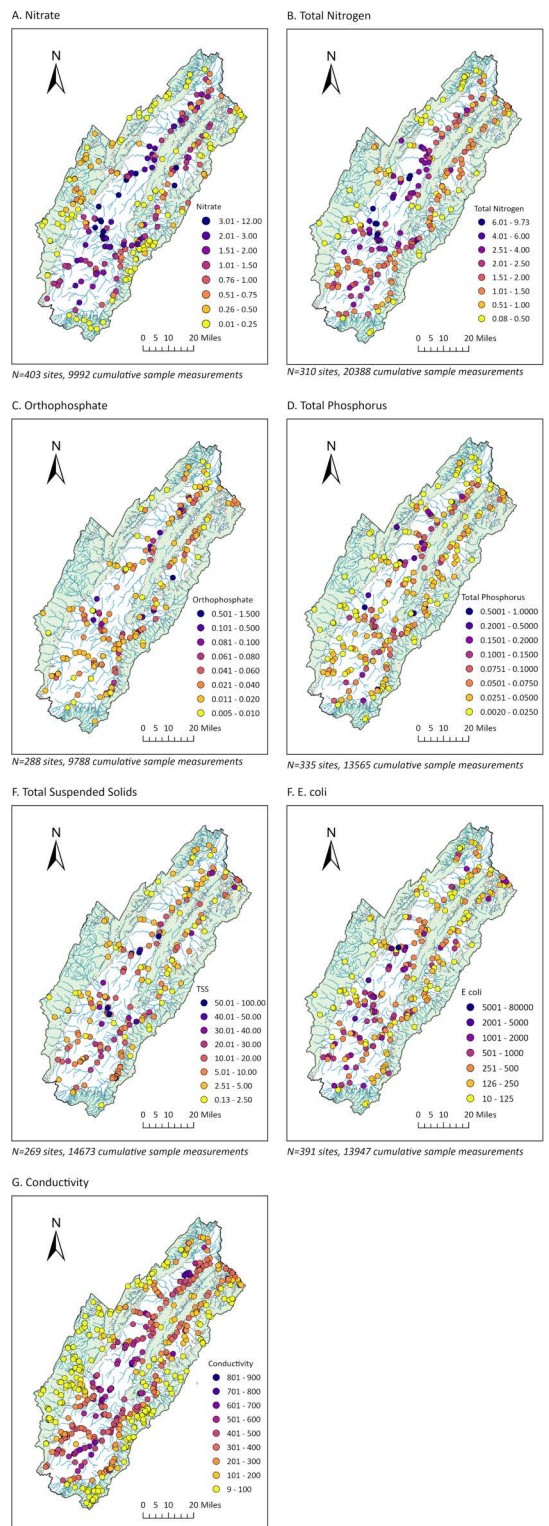

**Fig 3. Water quality parameters in watershed North and South Forks of Shenandoah River of Virginia.** Data are averages for sample sites for 2004-2023. Sample sizes are given as number of sites, and the total number of samples represented cumulatively at those sites.

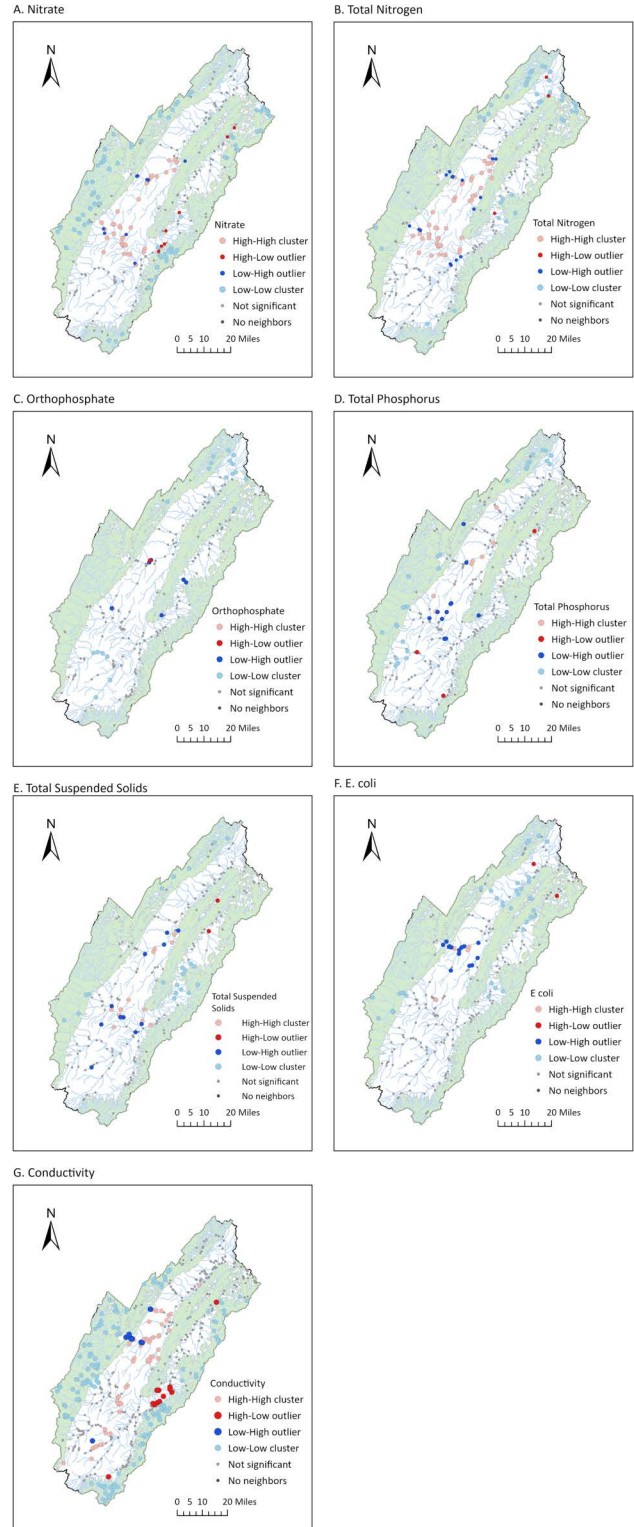

**Fig 4. Cluster and outlier analysis of water quality parameters for North and South Fork Shenandoah River, Virginia.** "High-High" and "Low-Low" are points in high and low cluster areas, respectively. "High-Low" and "Low-High" are high values in low cluster areas, and low values in high cluster areas, respectively.

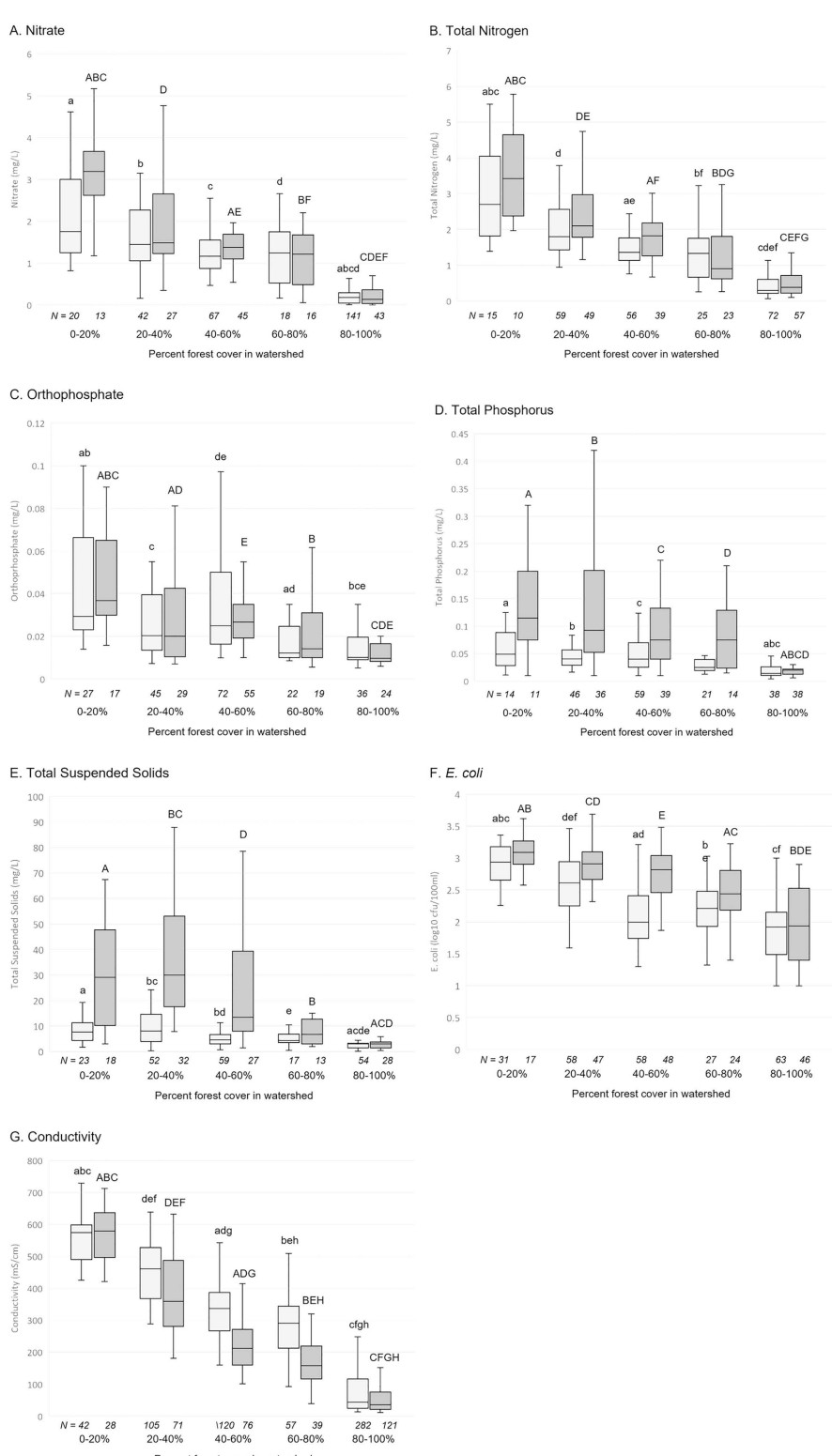

**Fig 5. Water quality in forest percent categories.** Light grey bars are for samples that were below the 75th percentile discharge volume, dark grey bars are for samples that were 75th percentile or above for discharge volume. Significant differences between categories indicated by letters on error bars. (lower case, p < 0.05 for lower 75th percentile of discharge volume; upper case, p < 0.05 for 75th percentile or above of discharge volume).

(Fig 5A, 5D and 5E). For other parameters, such as TN, orthophosphate and *E. coli* (Fig 5B, 5C and 5F), there was a somewhat more gradual decrease with increased forest cover, although there was still notably improved water quality at the highest forest cover categories. Conductivity showed the most gradual pattern of decreased concentrations with forest cover (Fig 5G). Overall, these results suggested that any effects of forestland on water quality extend into nonforested land, that the greatest decrease in water quality for these parameters occurred when rivers moved from heavily forested land to moderate or low forest cover, and that there is significant localized variation in all parameters.

Comparison of low and high flow conditions indicated that high flow mobilized TP, TSS, and *E. coli*. This was expected, as TP and *E. coli* are known to be associated with sediments, and high flow rates would mobilize sediments through processes such as bank erosion. Conversely, but also unsurprisingly, conductivity decreased in high flow conditions. The effects of high flow were absent or diminished at the higher forest cover categories, consistent with the protective role for forests in buffering flow rates and preventing erosion.

### Water quality in transects of three rivers flowing from forest to agricultural bottomland

We measured water quality along transects for three specific rivers – the Dry River, Briery Branch, and North River – located on the western side of the Shenandoah watershed. All three rivers flow from significant distances through national forest land, and then enter areas primarily used for agriculture (as both pasture and cropland), allowing us to explore more how water quality changes with the transition from forest to agriculture (Fig 2).

There was an increase in orthophosphate, nitrate, and general and fecal coliforms (Fig 6A, 6B, 6D and 6E) at 8–10 km downstream of the forest boundary. This pattern was less obvious for TSS (Fig 6C), which varied more at sample sites, and had some higher concentrations in the forestland. When we examined temperature differentials (how much temperature differed from the average temperature across the transect for that sampling run), we found a different pattern – temperature appeared to more gradually increase rather showing the abrupt shift at 8 river kms (S1 Fig). There was significant variation for all variables, indicating (as did the previous analysis of sites across the Shenandoah watershed; Figs 3 and 4) that there are significant localized effects that impact water at any particular point and time.

Conductivity also showed a clear change after rivers moved from forest to agricultural lands (Fig 6F), with the increase also happening about 8–10 river kilometers downstream from the forest edge. The increase was characterized by larger variability in the valley compared to the forested region. This increase corresponds to when sample site watersheds began to represent a more significant area of agricultural land (although the total area in watersheds was still mostly forestland) (open circles, Fig 6F).

Since there was an abrupt shift in some parameters at the 8–10 km mark, we pooled samples from the transects into three categories: forestland, edge of valley (up to 8 river kms), and interior of the valley (beyond 8 river kms). Given the known impacts of flow rate on water quality parameters, we furthered binned samples into three flow rate categories (low, medium and high). Two-way ANOVA did not show any effects of flow rate (p > 0.05 for all parameters). We therefore dropped the flow rate bins, and used one-way ANOVA with post-hoc to investigate the effect of location. For nitrate, orthophosphate, and general and fecal coliforms, forestland and edge of valley generally did not statistically differ from each other, whereas a significant shift occurred once water was in the interior of the valley (>8 km downstream) (Fig 7). This pattern did not hold for sediments.

We note that the 8 km mark, where percent of overall watershed generally drops to less than 80%, corresponds with the previous analysis of all Shenandoah valley sites indicating the biggest change in parameters likewise occurring with this change from heavily forested to moderately forested watersheds (Fig 5). The water quality benefits of forested land continue for some distance into agricultural valley land, but this likely depended on there being a large area of forest in the drainage basin (thus the overall watershed was still mostly forest). If there isn't a large area forest (such as some of the watersheds in the valley represented by sampling points in the overall Shenandoah watershed analysis), there are sites of higher water quality, but they are apparently determined by other factors besides forest cover.

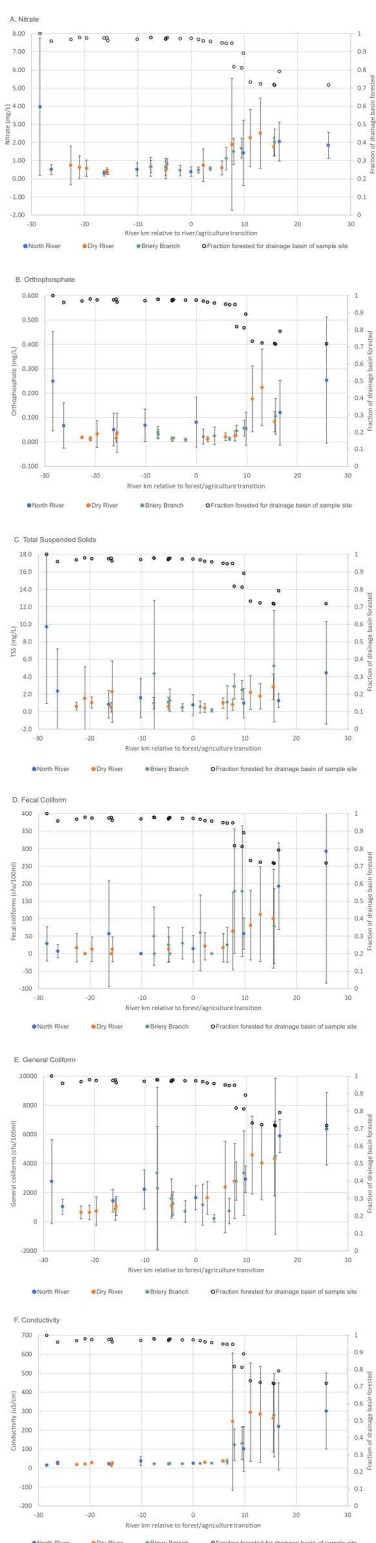

**Fig 6. Water quality parameters along river transects, from forest headwaters to agricultural bottomland.** Blue, orange and green points represent parameter mean±standard deviation values for the North River, Dry River, and Briary Branch, respectively. Open points represent percent forest fraction for each corresponding parameter point.

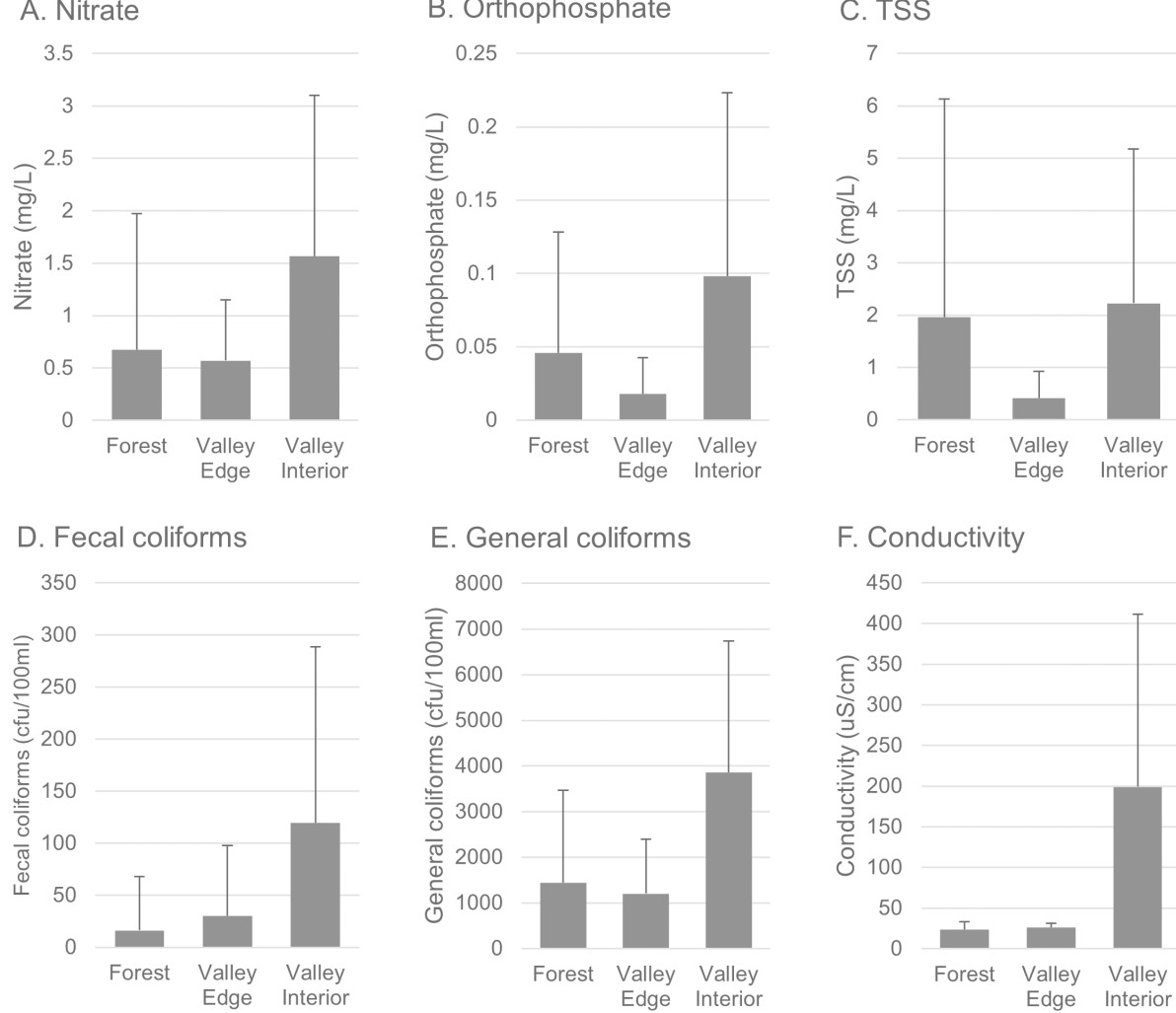

**Fig 7. Binned water quality parameters from transects.** Data is delineated by sites in forestland, sites within 8 km river miles downstream from the forest/valley transition (valley edge), and sites more than 8 km downstream of the transition (valley interior). Values are mean ± standard deviation; significant differences between categories indicated by letters on error bars ($p < 0.05$).

Finally, although sediment, nitrate and orthophosphate were lower in mountain regions, there were still levels of these parameters (and coliforms) that would be considered impaired, indicating forested regions are still a source of these pollutants at times. Orthophosphate and nitrate were in the medium to high stress level (0.05 ppm for phosphate, and 1 ppm for TN) in 17.6% and 13.5% of samples, respectively, at forest sites. Comparable percentages in the interior of valley samples were 45.2% of orthophosphate and 61.0% of nitrate.

## Discussion

The current results confirm what is generally understood about water quality in naturally vegetated (in this case forested) compared to agricultural areas – water quality is consistently better in watersheds which are well covered with natural vegetation such as forests [12], and agricultural lands can have water quality impacts [23]. It is evident, however, that within this general trend is important variability that occurs at localized areas. This variability is important not only for local stream health, but also collectively drives downstream impacts. Further, many waterways transect both forested and

agricultural areas, and the interaction between these two is less well characterized. By looking at a large public dataset, and focusing in on several representative watersheds, this study identifies some distinct patterns of water quality in a large watershed with distinct forest and agricultural segments.

## Key water quality parameters in forested vs agricultural land

When analyzed across the entire Shenandoah watershed, there was a clear pattern of forested vs agricultural land occurred for all parameters. Nutrients are key drivers of the downstream health of aquatic ecosystems, and in this case are central to the major efforts to restore Chesapeake Bay health. Phosphorus and nitrogen are both a focus of management efforts in upstream watersheds such as the Shenandoah. In fact, the Shenandoah/Potomac watershed system is one of the most impacted in Virginia for these nutrients [24]. In our study, both transect sampling and analysis of nutrients in the water quality portal database indicated a general pattern of higher nutrient levels in agricultural regions that clearly agrees with the well-studied potential of agricultural practices to impact water quality through anthropogenic activities. We did not explore specific drivers of nutrient dynamics in the agricultural area, but note that the relationship between agricultural practices and nutrient levels is complex. For instance, the large scale analysis of Metson et al. [25] found that there was no single explanatory variable that determines riverine phosphorus levels, and other studies have found variable impact of best management practices [16].

We note here four observations on nutrient trends from our data. First, our results are roughly consistent with those found in other studies that focused on this region, both in average concentrations [24,26,27], and in the presence of significant variation on the local level [15]. Second, many of these values were above estimated threshold criteria for algal and benthic macroinvertebrate impairment [28], and are outside of the EPA suggested criteria for ecological health [29]. Mean nutrient values for mountain and valley regions were in the low and medium categories for probability of stress to aquatic organisms, respectively, according to Virginia stressor analysis [24]. Third, values in areas of low impact (forested regions in the mountains) also reached levels of concern at times, consistent with Smith et al [15] who found large variation in natural background of phosphorus and nitrogen, with many stream and river segments in this ecoregion exceeding EPA-proposed criteria. Fourth, phosphorus (orthophosphate and TP) were more variable in the valley than nitrogen (nitrate and TN) – nitrogen was consistently high in the valley, whereas phosphorus was high at some sites, and low at others. Although our data do not indicate what governs this difference (e.g., source, transport, adsorption, etc), it is consistent with the observation in another study that TP has a smaller patch size than TN [5].

Two other measured parameters – sediment (as total suspended solids) and bacteria – are of major concern nationwide as drivers of water quality, and have clear associations with agricultural activity. For instance, sediment is the largest cause of invertebrate-impaired waters nationwide [30] and the primary stressor in the Chesapeake Bay as a whole [8], and bacteria is the most common reason for streams receiving an impairment designation in Virginia [24]. These parameters also showed some patterns of higher levels in agricultural lands, although the pattern was more variable than for nitrogen and conductivity. For instance, sediments (TSS) were highly variable in the transect data and did not correlate with land cover, but TSS across the watershed was higher in drainage basins with less forest cover. Bacteria did correlate with land cover, but generally showed more variability in both the transect and total watershed data. Overall, this points to some higher sediment and bacteria values in forested land (see discussion below), and/or some waterways in primarily agricultural regions that are less impacted by sediments and bacteria. Thus, although forest cover and agricultural activities are critical for water quality, other factors are driving significant local variation in these parameters.

Conductivity was also clearly different between mountains and valley, and showed a more linear gradient with change in forest cover. Conductivity was the single parameter that showed a correlation with multiple potential drivers (Table 1) – at high significance for both forest cover and erodibility, and moderate significance ($p < 0.1$) for karst. It is likely, therefore, that some of this pattern is driven by the change from silicoclastic to karst geology as rivers move from forest to valley. Although there are human activities of concern for conductivity (such as road salting, or in areas with industry or mining

[Fanelli]), studies in nearby Appalachian mountain areas point towards agricultural having relatively low effect on conductivity [31], and conductivity is the parameter of least concern as a stressor in the region [4,24]. In all cases, the mean conductivity was below 300 μS/cm, generally recognized as the range of no or low probability for stress to aquatic life [24,29,30]. However, some individual samples fell in the high stressor categories (>500 μS/cm), and could be a driver of which organisms make up aquatic communities in these areas. These high conductivity values occurred during low-flow periods in valley areas only, usually during mid-summer.

## Water quality impacts in forested land

Although water quality was better in forested land, our data indicated that nitrogen, phosphorus, TSS and indicator bacteria all exceeded levels of concern at particular times and locations. These results are consistent with Bricker et al [14] who likewise found similar nitrate levels in a nearby mountain area, and with Price and Leigh [32] who found TSS levels in forested areas of the southern Appalachians as high or higher than the levels throughout our study area. Smith et al's [15] nationwide study also found some sites and times with levels of concern in relatively unimpacted areas of our ecoregion.

We did not specifically investigate the source of these pollutant levels in forestlands, although our observations pointed towards three general categories of possible human-related contributors: 1) existing stable structures, 2) various types of disturbance, and 3) legacy effects. The contribution of existing stable structures is indicated by consistently high levels of measured parameters (except conductivity) in retention structures such as a small pond located in a clearing at the headwater of the North River. This pond was in a small sedimented bowl on the ridgeline, and is characteristic of numerous small retention basins purposefully maintained in the forest (usually for supporting game animals). Larger reservoirs are also present in all three watersheds for the purpose of flood control and municipal water supplies. Recent evidence suggests that ponds can serve as nutrient sinks in agricultural areas [33]; to our knowledge, there are no similar studies on the water quality role of ponds or lakes in forested areas.

Disturbance is recognized as critical for mobilization of nutrients and sediments. For instance, forests in this area are considered to show low nitrogen export unless disturbed [34]. While heavily forested, there are a variety of processes which impact forests in a way that could mobilize nutrients and sediments. These regions have a network of roads which are used for recreation, forest service activities (*e.g.,* prescribed burns or firefighting), and resource extraction (*e.g.,* timbering). These roads often parallel waterways, are usually unpaved, and in some areas have low-water crossings rather than bridges. Roads are a small fraction of the total area, but potentially a significant contributor to water quality, and deserve future attention to understand their role forested regions. Timber harvesting and prescribed burns are regular localized activities that can have significant impacts on streams. In addition, our observation is that some streams in the forest areas have eroded banks, either part of natural processes or as a legacy of alterations previously made, which would be a likely source of nutrients and sediment. Bare soil from both management activities or streambank erosion are known drivers of water quality and quantity in forested areas [10,35], and thus are likely contributors to the occasionally high levels of nutrients and sediments in our study.

Candidate drivers of some water quality parameters include not only present day activities, but also legacy activities. In addition to management activities occurring over the course of many decades, this area was heavily logged in the late 1800's, and then used for livestock grazing before forests were restored in the early 1900's. The role of legacy nutrients as drivers of watershed nutrients has received increased attention as a key driver of water quality (*e.g.,* [36]) as studies demonstrate their importance in a variety of watersheds [37–39]. Legacy nutrients probably have variable residence times depending on local conditions, but are likely on the order of many decades [39,40]. Mobilization from soil is particularly important during storm events following freeze-thaw cycles [38], such as the spring rains often experienced in this central Appalachian region. Although these observations all point toward a potential role of legacy nutrients in forested land, at present there is no direct data to guide an understanding of how much legacy nutrients currently contribute to elevated nutrients in this area.

Finally, other "natural" processes can impact water quality in less disturbed watersheds. Atmospheric deposition is shown to be important for both nitrogen [15] and phosphorus [6] in relatively undisturbed landscapes. Insect outbreaks can dramatically increase nitrate from vegetation inputs, as shown in studies from Shenandoah National Park, which borders the eastern side of this watershed [34]. Wildlife have been shown to be significant sources of fecal coliforms in watershed analyses [41], and are likely a significant source of this parameter in our forested areas.

In summary, although a national survey of landscapes found forests with the highest water quality among all land cover types [12], a narrative that waterways in forested landscapes as invariably "clean" masks some interesting localized events, and should be distinguished from the general observation that waterways are cleaner on average. In fact, the recent study identifying forests as having the best water quality also emphasized considerable variation in sediment and nutrient levels [12]. Forestlands in this region are a complex amalgam of natural and human processes – as would be the case for most forests in the United States (and indeed globally). Further studies should extend the general forest vs non-forest paradigm to focus on what categories of activities in forests impact water quality more or less.

## Patterns of water quality transition as rivers flow from forested to agricultural lands

Although this area is notable for a sharp forest/agriculture transition, transect samples indicated water which flowed out of forested land did not show an immediate change in water quality parameters. Rather, there was a clear lag period as the water quality parameters remained relatively unchanged until about 8 river km downstream of the forest boundary (Figs 6 and 7). Although residence time (in this case, time in an agricultural area) is a known driver of water quality [23], residence alone would more likely produce a gradual degradation of water quality. Rather, the decrease in water quality was fairly abrupt, most notably for conductivity, which tracks with multiple factors including underlying geology which shifts at the valley edge. When percent forest area of watersheds for each sampling point were calculated, forested area for the corresponding drainage basins also showed a sharp decrease at the 8 km mark to about 70–80%. WQP database results were similar, with many parameters showing the most significant drop in quality when forested areas dropped below 80% (Fig 5).

The 8 km mark is the point at which tributaries draining mostly agricultural areas become a larger part of the total drainage basin. In turn, this suggests that cumulative inputs from agriculturally derived tributaries are not substantial until the river is 8 km into the agricultural area, and that the pattern may be driven by inputs from the broader tributary watershed rather than gradual inputs from agricultural activities immediately adjacent to the river. This is consistent with the observation that sediment loads are correlated with bank erosion, rather than primarily with runoff from adjacent land. Thus, it seems that the first 8–10 km remains unimpacted because the drainage area from non-forested regions is low. Once tributaries join, or substantial subsurface water from the valley joins, the water quality begins to drop. Furthermore, while the benefits of the forested region stretched well into the agricultural region, it took only 20–30% of a watershed to be nonforested in order for water quality to be impacted. Price and Leigh [32] likewise found that even moderate changes in the degree of forestation can have impacts on water quality variables. In aggregate, these data imply that 1) the value of forest in preserving water quality is substantial, and extends downstream out of the forested area, but 2) that the ecological service of maintaining water quality drops off once 20–30% of a total watershed was agricultural.

Although we did not focus on temperature, it is a parameter of critical concern in the area, as long-term measurements in Shenandoah National Park show an increase that is expected in the presence of climate change [42]. We did measure water temperature, but the interpretation of single temperature readings taken at different times of the day and season limits conclusions from the transects. However, a rough estimate of transect trends can be estimated by standardizing individual readings as the deviation from the average of a sampling run along a transect. Temperature showed a more gradual increase across the length of the transect. Temperature steadily rose as water moves from headwaters to the end of the sampling transect (S1 Fig), contrasting with the abrupt transitions seen in other parameters. The slow increase would be expected for temperature which is driven by complex energy exchanges that rely on atmospheric factors (such as time with exposure to sun) rather than land cover or other surface/subsurface factors [43].

Two other factors are key at governing the dynamics of water quality parameters, and are relevant to this watershed – underground flow and storm events. Although we did not study the dynamics of underground flow, which would require additional methodology such as flow tracing, others have pointed out the importance of subsurface flow in this region [44,45]. The valley area is predominantly (but not entirely) karst, where underground flow would be particularly important for impacting water quality. For instance, two of our transect rivers are known to be interconnected – the Dry River flows underground into a tributary (Beaver Creek) of Briery Branch [44]. This highlights how water quality at any point in fact likely represents a cumulative impact beyond the drainage basin itself. Futhermore, karst may govern contaminants in other ways, such as through the adsorption of TP [46]. Underground processes could be crucial to management efforts in systems such as Smith Creek, an extensively studied watershed in our study area where the contribution of underground discharge is greater than rainfall runoff [45]. For instance, mitigation interventions such as vegetated buffers along streams may be less effective at intercepting contaminant inputs if water bypasses the buffer through underground flow.

High flow rates produced by storm events can drive the largest portion of loads [17,47,48], as demonstrated for specific watersheds in the Chesapeake Bay system [45]. That study found that 90% of the sediment load in Smith Creek in a three year period was due to eight large storms. In addition to increased streambank erosion in headwaters, storm events initiate flow in the large number of as ephemeral streams, which mobilizes previously stable pollutants [49]. We performed an initial investigation on the impact of storms by categorizing WQP data generally as high and low flow, and we highlight several specific insights from our data that contribute to an understanding the role of storms in water quality: 1) The effects of high flow varied with parameter. For instance, phosphorus was mobilized more than nitrogen, which is consistent with observations in other studies [45], and has implications for the impacts of high nutrient levels on algae growth [50]. Bacteria and sediment were also mobilized, whereas conductivity decreased. Although nitrogen concentrations did not increase, the increased flow rates would result in an increased nitrogen load, but to a lower degree than for phosphorus, bacteria and sediment (where both concentration and flow rate increased). 2) The effects of high flow varied with forest coverage. Notably, periods of high flow where forest coverage was high (>80%) did not cause the increased phosphorus, bacteria and sediment concentrations noted above. Thus, heavily forest areas moderated the increases in pollutant loads that occur during storm events, whereas drainage basins with more moderate or low forest levels had higher loads through increased contaminant concentrations. Seasonal impacts would further complicate these dynamics, as changes in management practices (such as manure or fertilizer application), and/or "natural" sources and sinks such as leaf drop and algae growth, change the pollutant source and sinks. These results highlight the continued need to understand the role of storm events on water quality.

## Conclusions

The current study confirms the crucial role of substantial forested areas regions to the water quality of watersheds. While the importance of headwater regions to downstream water quality is recognized [17,51], this study combined multiple datasets to explore how water quality changes with the transition of waterways from forested areas, which generally protect water quality, to agricultural areas, which generally degrade water quality. Key insights which add detail to the general understanding of these landscapes are: 1) substantial variability in water quality in both agricultural and forest areas points towards important local factors governing pollutants, 2) while forests clearly had higher water quality than agricultural areas, a narrative of pristine water in forested areas does not reflect the reality of parameters reaching levels of concern at certain locations and times, and 3) forested regions provide benefits substantial distances downstream of transitions to agricultural regions, but water quality rapidly decreased when cumulative forest cover of a corresponding watershed dropped by 20–30%. These results contribute to a general understanding of what drives water quality in mosaiced landscapes such as the Chesapeake Bay watershed, which has shown complex and variable responses to management efforts [7].

While the current study helps draw a more complete picture of water quality across a diverse landscape, we did not investigate all possible drivers of water quality. Several other factors that are critical to understanding when and how

degradation of water quality occurs – notably the contribution of underground flow, storm events and seasonal cycles. Flow rates, for instance, are a known and crucial determinant of loading [17,47,48], and nutrients can follow seasonal trends [14]. Preisendanz et al [52] argue that "hot moments" are as important as "hot spots" – and that insufficient attention has been directed towards these temporal events as opposed to the spatial pattern. Whether the patterns seen here represent those found during peak storm flow, and across different seasons, is a further avenue of fruitful study, particularly in light of future climate changes [53]. Future increases in precipitation are highly relevant given that downstream water quality in the Chesapeake Bay is tied closely to flow rates [17], and recent estimates are that climate change threatens to increase nutrient and sediment loads in the Chesapeake Bay watershed even more than changes in agricultural intensity, population or development [54], as nonpoint sources become more important [55].

## Supporting information

**S1 Table. Dates and flow categories of transect sampling events.**
(DOCX)

**S1 Fig. Temperature differential gradient parameters along river transects, from forest headwaters to agricultural bottomland.** Temperature differential for each sample calculated as site temperature minus average temperature across its run (samples for the one river and one date). Blue, orange and green points represent parameter mean ± standard deviation values for the North River, Dry River, and Briary Branch, respectively. Asterisks with higher temperature differentials are pond or reservoirs.
(TIF)

## Author contributions

**Conceptualization:** Douglas Neufeld.

**Data curation:** Douglas Neufeld, Isaac Alderfer, Zachary Bauman, Micah Buckwalter.

**Formal analysis:** Douglas Neufeld.

**Funding acquisition:** Douglas Neufeld.

**Investigation:** Douglas Neufeld, Isaac Alderfer, Zachary Bauman, Micah Buckwalter.

**Methodology:** Douglas Neufeld.

**Project administration:** Douglas Neufeld.

**Resources:** Douglas Neufeld.

**Supervision:** Douglas Neufeld.

**Validation:** Douglas Neufeld.

**Visualization:** Douglas Neufeld.

**Writing – original draft:** Douglas Neufeld.

**Writing – review & editing:** Douglas Neufeld.

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
