## [Decision Letter · Decision Letter 0]

13 Feb 2025

Dear Dr. Neufeld,

Thank you for submitting your manuscript to PLOS ONE. After careful consideration, we feel that it has merit but does not fully meet PLOS ONE’s publication criteria as it currently stands. Therefore, we invite you to submit a revised version of the manuscript that addresses the points raised during the review process.

We look forward to receiving your revised manuscript.

Kind regards,

Gurpal S. Toor, Ph.D.

Academic Editor

PLOS ONE

Journal Requirements:

1. Please ensure that your manuscript meets PLOS ONE's style requirements, including those for file naming. The PLOS ONE style templates can be found at https://journals.plos.org/plosone/s/file?id=wjVg/PLOSOne_formatting_sample_main_body.pdf and  https://journals.plos.org/plosone/s/file?id=ba62/PLOSOne_formatting_sample_title_authors_affiliations.pdf.

2. Please include a caption for table 1. 

 [The authors gratefully acknowledge funding received internally at Eastern Mennonite University.].  

5. We notice that your supplementary figures is included in the manuscript file. Please remove them and upload them with the file type 'Supporting Information'. Please ensure that each Supporting Information file has a legend listed in the manuscript after the references list.

Reviewers' comments:

Reviewer's Responses to Questions

**Comments to the Author**

1. Is the manuscript technically sound, and do the data support the conclusions?

Reviewer #1: Partly

Reviewer #2: Yes

2. Has the statistical analysis been performed appropriately and rigorously?

Reviewer #1: No

Reviewer #2: Yes

3. Have the authors made all data underlying the findings in their manuscript fully available?

Reviewer #1: No

Reviewer #2: Yes

4. Is the manuscript presented in an intelligible fashion and written in standard English?

Reviewer #1: Yes

Reviewer #2: Yes

Reviewer #1: With catchment-scale data points over 20 years and transect data points over four years, this study investigated the impacts land use on water quality in the Shenandoah River watershed. The results showed that the water quality in the forested areas is better than that in the agricultural areas, and a distance of 10 km after the transition from forestland to agricultural land was identified to be critical for water quality change. I can imagine the difficulty in field sampling and data analysis since the environment is too variable. But the manuscript can be improved by incorporating more public data and methods.

Major Concern:

(1) The data and method section should be improved to give more details to increase the reliability of data.

(2) Although the water quality data were grouped through land use pattern or distance after the transition from forestland to agricultural land, the deduction appears not persuasive since the analysis ignores too many environmental factors. For example, the soil would have great impacts on the water quality. Although the soil conditions can be reflected by land use pattern if each point has consistent land use type, the transition region may not be the case since they are influenced by more complicated conditions. It is better to incorporate more environmental information.

(3) After incorporating more environmental information, some analysis can be conducted to identify the dominant factors controlling water quality, such as geospatial detector, random forest methods. It the identification directly show that land use has the most important impacts, it would be better than current deduction.

Introduction

This section is overall simple with qualitative description. It can be expanded to give more analysis and summarization. In particular, some quantified results from different studies over a large spatial scale should be presented. The research questions, summarized from literature review, should be further highlighted in different paragraphs and the last paragraph to clarify the novelty of this study.

Methods

(1) Line 80. How to access the water quality portal. Website or other information should be presented.

(2) Line 86. It is better to present more details for the number of samples, in the form of table or figure. The information is very important.

(3) Line 87-89. What caused the outliers? Have you analyzed the time or sites of these outliers? Is there any mechanism or reason?

(4) Line 94. The authors only used the 2021 land use data to represent the land use pattern over 20 years. I don’t think this is right if the catchment has significant land use change in the past 20 years. The land use change condition should be examined first, and land use map for different years should be used if there is significant land use change.

(5) Line 104. Figures should be presented to show the distribution of sampling rivers and sites, as well as the basic information such as river network, DEM, land use pattern, soil, climate.

(6) Line 115. ‘Samples were collected between March 2020 and July 2024 and were taken at all times of the year.’ What is the sampling frequency?

(7) The methods for data analysis should be presented. The current description focuses on sample collection and determination, which is far from enough.

Results

(1) It is better to use the same legend for the samples or sampling sites for the three rivers shown in Figure 5 and 6.

(2) With 10 km as critical distance, where would the sampling sites be in Figure 5? The river networks appear different after they enters in the agricultural regions, why they have the same critical distance for water quality change? If it is not related to water from different tributaries, it may be related to the soil?

Reviewer #2: General Comments:

Overall, the manuscript titled “Headwaters to Valley: Water Quality in Rivers Transitioning from Forest to Agricultural Bottomland” was very well written and serves as a large data set paper to help drive home important points about watershed management and land-use impacts on water quality. I do have some specific comments below and a few general comments in this paragraph that I think could help make this manuscript stronger. There was no seasonality breakdown in the 20 years of water quality parameters. Was any work on separating the measurements based on season (wet spring vs dry summers) done? I couldn’t help but think about this when reading the low-high and high-low outliers section. Additionally, when discussing the three rivers that were sampled via transects, the water quality data seemed lumped throughout. However, when assessing the map (Figure 5), it seemed like the Dry River potentially had more development next to the river. Was this true? Did this make the Dry River’s water quality data different from the other two rivers? Throughout this section I was constantly thinking that it would be nice to include a section that compared the three rivers to see if there were any differences (as there inevitably will be on occasions). Once again, I do think this manuscript is well written and takes into account forests’ impact on watershed water quality parameters. All comments written here should be taken as minor as their additions wouldn’t change the overall perspective of the manuscript.

Specific Comments:

Lines 56-59: There has been little talk about karst topography could be a factor within these limestone areas. These potentially large flow pathways can have a large effect on nutrient, sediment, and e coli analysis. Consider addressing this throughout the paper.

Lines 85-89: Since you have so many water quality parameters with varying number of samples, outliers, etc, it may be nice to include a table with these values. You can either put it in the main text, or consider adding a supplemental section for the table.

Line 98: How was the 75% coverage decided upon? Is this based on other papers, or was it developed here based on trial and error of assessing the watershed maps? Please explain this further as future studies that could potentially do similar methods would need to know.

Lines 104-109: This is all really hard to visualize for people not familiar with the watershed. Consider adding a map that highlights the different sections of the river (without the water quality dots on it like Figure 1).

Lines 111-119: May be good to put some of this on the map mentioned above for Lines 104-109. The statement “river sections in the middle interior of the valley dried completely during periods of low rainfall” before the “varied from 114 to 1430 ft3” seems counter intuitive since the reader is not aware of these locations.

Line 159: Where was the landcover data from? Said it was documented, but does the WQP data have that kind of detail over time? If so, how often was landcover assessed?

Lines 160-162 and Figure 1: Are these average concentrations over the 20 year dataset? Please specify this in this paragraph and on Figure 1’s caption.

Lines 197-208: The outliers, this large data set, and the talk about changing stream flow conditions all has me wondering about seasonality. I am sure with this large data set, that analysis could be made for changes in this forestland theory in the wetter (larger flow) months vs how it compares in drier (smaller flow) months. There are many watershed papers out there that have seen nutrients and sediments respond to seasonality, and even more so after precipitation ends a long drought season. Consider doing some of this analysis to assess whether the outliers (either low-high or high-low) can be stemming from seasonality.

Lines 224-231: There is a good deal of thoughts from this section. Here are a few that really stood out in my mind when reading this section. Much of this analysis comes from focusing on percent forested land, but I am also wondering about the percent of agricultural land once the streams leave the forest. Does this change depend on what river you are sampling? Overall, how did the three rivers’ results differ from one another? It almost seems like the Dry River in Figure 5 has more development (red) areas along the stream, even near the orange circle that is surrounded by forest. Were TSS samples different between the rivers, especially the Dry vs the Briery Branch and North River due to differences in development?

Figure 7: I do think that the high TSS values in the forested land should be looked into further. Why is this? What happens to all the sediment before the water reaches the Valley Edge? Are there settling ponds/beaver dams/wetlands/etc? Where is this sediment coming from? Do all three rivers have higher TSS. The way all the data is aggregated into one analysis makes picking apart trends between rivers (and their subsequent land use differences) difficult. Also, what if one was to assess TP instead of orthophosphate? Would the TSS also lead to larger TP loads in the forested portion of the watershed? (I know you didn’t measure TP in this study, but the hope is that you are able to mention this possible TP and sediment link in this study for future similar watershed studies.

Lines 364-365: Do the number of roads differ between rivers, and is that correlated to the TSS levels in each?

**Do you want your identity to be public for this peer review?** For information about this choice, including consent withdrawal, please see our Privacy Policy

Reviewer #1: No

Reviewer #2: No

---

## [Author Response · Author response to Decision Letter 1]

14 Jul 2025

Thank you for the reviewer comments and suggestions on our manuscript. Please find the revised manuscript now submitted, with changes and/or responses to suggested changes. Our sense from the comments was that the premise of the study was good and that the writing was good, but there needed clarity on some presentation, and additional analysis. We note here considerable efforts to address these requests. In this document, please note that comments from the journal or reviewers are in black text;.we’ve indicated our responses in blue text in the attached "response to reviewers" letter, and lines with our responses here start with an asterisk (*).

I would highlight several general ways in which we’ve addressed major suggestions:

- Request to be more clear on protocols: We’ve increased both the analysis and documentation of methods to address any concerns about “technical soundness” or “statistical analysis” of the data. We’ve uploaded to Zenodo additional worksheets with all data and steps.

- Request to perform additional analyses: Since both reviewers suggested additional analyses, we have added those. Note that we made what we felt was a strategic decision to narrow in on the bigger picture conclusions, rather than doing too many analyses that increased the length of the paper, and risked diluting the impact of the main conclusions.

- Clarity/content on introduction and discussion: There were suggestions to extend the text to more comprehensively set the context, and interpret the results. While we generally want to keep the text very focused and avoid making this a review of many different topics, we agree that more text increases some areas of clarity. We’ve thus added some sections.

Thank you for your patience as we made revisions to this manuscript. The extra time for revision was in order to carefully address the substantial requests of reviewers for more analysis and description. We note that we also discovered an error in some of the analysis of the water quality portal data (Unbeknowst to us, Excel had defaulted to assigning water quality parameters to some sample points by approximating the correct label. In other words, it inappropriately assigned water values to some points). This error was a straightforward fix, but we spent substantial time reviewing all data to make sure there were not additional errors. We added additional spreadsheets to zenodo with all steps of the data analysis, so that values can be verified by anybody. Correction of the error changed some individual values, but did not change the overall conclusions of the paper - the trends remain the same with the corrected values. We regret the initial error. We are confident that the values are now correct, and we’re happy to answer any questions about this.

Sincerely, Doug Graber Neufeld

Response to journal requirements

Journal Requirements:

1. Please ensure that your manuscript meets PLOS ONE's style requirements, including those for file naming. The PLOS ONE style templates can be found at https://journals.plos.org/plosone/s/file?id=wjVg/PLOSOne_formatting_sample_main_body.pdf and https://journals.plos.org/plosone/s/file?id=ba62/PLOSOne_formatting_sample_title_authors_affiliations.pdf.

*The style is changed to match that of PLOSOne. We’d originally submitted this in the “style-free” format; all formatting is now changed.

2. Please include a caption for table 1.

*All captions are now added; apologies that the caption was omitted.

*This has been corrected.

[The authors gratefully acknowledge funding received internally at Eastern Mennonite University.].

*Thank you for the clarification, we’re including the suggested statement in the cover letter.

5. We notice that your supplementary figures is included in the manuscript file. Please remove them and upload them with the file type 'Supporting Information'. Please ensure that each Supporting Information file has a legend listed in the manuscript after the references list.

*This has been corrected.

Responses to reviewers comments

Reviewer's Responses to Questions

Comments to the Author

1. Is the manuscript technically sound, and do the data support the conclusions?

Reviewer #1: Partly

Reviewer #2: Yes

2. Has the statistical analysis been performed appropriately and rigorously?

Reviewer #1: No

Reviewer #2: Yes

*Additional information is indicated on the statistical analyses applied. We trust that this satisfies the request of reviewer #1

3. Have the authors made all data underlying the findings in their manuscript fully available?

Reviewer #1: No

Reviewer #2: Yes

*Additional spreadsheets are made available. Originally we had included the transect data (our measurements) available in zenodo, and pointed to the Water Quality Portal for the other data. As per the concern of reviewer #1, we’ve now included the WQP data that we used on its own spreadsheet, plus added spreadsheets for the various analyses. All steps from the original data through analyses are now available on zenodo. ________________________________________

4. Is the manuscript presented in an intelligible fashion and written in standard English?

Reviewer #1: Yes

Reviewer #2: Yes

5. Review Comments to the Author

Reviewer #1: With catchment-scale data points over 20 years and transect data points over four years, this study investigated the impacts land use on water quality in the Shenandoah River watershed. The results showed that the water quality in the forested areas is better than that in the agricultural areas, and a distance of 10 km after the transition from forestland to agricultural land was identified to be critical for water quality change. I can imagine the difficulty in field sampling and data analysis since the environment is too variable. But the manuscript can be improved by incorporating more public data and methods.

*By “...incorporating more public data and methods”, we assume the reviewer is suggesting more clarity in how we used the data we worked with. We have expanded the description of our methods in the text, made multiple spreadsheets now available with the data and analysis steps.

Major Concern:

(1) The data and method section should be improved to give more details to increase the reliability of data.

*We’ve given more details for both the data used, and the methods. As one example, we added a paragraph clarifying how and why outliers were handled the way they were, which was a specific question by a reviewer.

(2) Although the water quality data were grouped through land use pattern or distance after the transition from forestland to agricultural land, the deduction appears not persuasive since the analysis ignores too many environmental factors. For example, the soil would have great impacts on the water quality. Although the soil conditions can be reflected by land use pattern if each point has consistent land use type, the transition region may not be the case since they are influenced by more complicated conditions. It is better to incorporate more environmental information.

*Indeed, this is a good point that there are various environmental variables that could drive water quality, and we do not want to overlook this. We’ve added an entirely new analysis to address this concern - performing a correlation of water quality not only with land cover, but also a soil characteristic (erodibility), and underlying geology (which would also be reflected in the soil). We chose these as they seemed the most likely to drive water quality variables, and there is good spatial data for analysis. We acknowledge there might be other candidate factors (for instance, climate variables, or long term historical land patterns), but suggest there is not a feasible way to test for every possible environmental factor. The factors we choose are arguably the best candidates, and helpfully point to a conclusion even if we are not able to analyze every possible factor in the scope of this study. Finally, we note that focusing on land cover is consistent with the many studies in the literature providing evidence for the key role of land cover - those studies are cited in our manuscript, such as in the introduction.

(3) After incorporating more environmental information, some analysis can be conducted to identify the dominant factors controlling water quality, such as geospatial detector, random forest methods. It the identification directly show that land use has the most important impacts, it would be better than current deduction.

*As indicated above, we have now analyzed multiple candidate drivers of water quality. Statistical analysis was then performed to show forest cover as the main driver, of those candidates tested. As such, this confirms our original conclusion, and we believe strengthens the rationales behind our conclusions.

Introduction

This section is overall simple with qualitative description. It can be expanded to give more analysis and summarization. In particular, some quantified results from different studies over a large spatial scale should be presented. The research questions, summarized from literature review, should be further highlighted in different paragraphs and the last paragraph to clarify the novelty of this study.

*We’ve added additional information here through a paragraph which focuses on several key studies, at the large spatial scale and at local scales. More could be added, but we believe that too much additional text reviewing many studies would detract from the specific focus of the study.

Methods

(1) Line 80. How to access the water quality portal. Website or other information should be presented.

*The WQP and FOSR websites have been added

(2) Line 86. It is better to present more details for the number of samples, in the form of table or figure. The information is very important.

*Sample size was added to individual figures in order to better present details on the number of samples. This seemed more effective than adding a separate table, as the reader can then see for each dataset how many samples were analyzed.

(3) Line 87-89. What caused the outliers? Have you analyzed the time or sites of these outliers? Is there any mechanism or reason?

*What caused the statistical outliers is unknown and probably varies. Being a large database drawing from multiple organizations, there is less certainty whether outliers would be measurement error, or actually anomalous amounts in the environment. Regardless, the method we use is a standard statistical technique for removing outliers that would bias the data. Further, we would argue it’s less critical to know what drives an outlier if the goal is to characterise the “normal” situation in these waterways (whether in baseflow or high water). What drives short temporal spikes in a water quality parameter, for instance, is a good question worth studying, but outside the scope of this study.

(4) Line 94. The authors only used the 2021 land use data to represent the land use pattern over 20 years. I don’t think this is right if the catchment has significant land use change in the past 20 years. The land use change condition should be examined first, and land use map for different years should be used if there is significant land use change.

*This is a very fair point, thanks to the reviewer for pointing this out. We’ve compared land cover from 2021 with 2003, and found there was relatively little change over this period (<1%) for forested vs unforested areas. Given the lack of significant changes, we stayed with the original 2021 land cover dataset for our analyses. This is indicated in the methods section.

(5) Line 104. Figures should be presented to show the distribution of sampling rivers and sites, as well as the basic information such as river network, DEM, land use pattern, soil, climate.

*We’ve added our modified figures to clarify additional features that would be of help to the reader (for instance, soil erodibility, and land cover)

(6) Line 115. ‘Samples were collected between March 2020 and July 2024 and were taken at all times of the year.’ What is the sampling frequency?

*Specific sampling times varied for logistical reasons, we’ve added dates to the supplementary data.

(7) The methods for data analysis should be presented. The current description focuses on sample collection and determination, which is far from enough.

*More detail on analysis methods is given.

Results

(1) It is better to use the same legend for the samples or sampling sites for the three rivers shown in Figure 5 and 6.

*The same symbols for each river are now used in Fig 5 and 6 (they were shown differently in the map vs graph).

(2) With 10 km as critical distance, where would the sampling sites be in Figure 5? The river networks appear different after they enters in the agricultural regions, why they have the same critical distance for water quality change? If it is not related to water from different tributaries, it may be related to the soil?

*Indeed individual rivers all have individual differences, however we would argue that, in broad strokes, these 3 river systems are more similar than different. We added specific land use data to figure 2 in order to show this. That data shows that (for instance) developed land is low in all 3 systems, and that forest land and pasture/crop land are dominant in similar ways. Undoubtedly there are interesting questions parsing out individual differences in rivers, however we are focusing on large patterns and similarities in this study. [See additional comments later in this response also.]

Reviewer #2: General Comments:

Overall, the manuscript titled “Headwaters to Valley: Water Quality in Rivers Transitioning from Forest to Agricultur

---

## [Editor Report · Decision Letter 1]

23 Jul 2025

Headwaters to Valley: Water quality in rivers transitioning from forest to agricultural bottomland

PONE-D-24-57524R1

Dear Dr. Neufeld,

We’re pleased to inform you that your manuscript has been judged scientifically suitable for publication and will be formally accepted for publication once it meets all outstanding technical requirements.

Congratulations!

Kind regards,

Gurpal S. Toor, Ph.D.

Academic Editor

PLOS ONE
---

## [Editor Report · Acceptance letter]

PONE-D-24-57524R1

PLOS ONE

Dear Dr. Neufeld,

I'm pleased to inform you that your manuscript has been deemed suitable for publication in PLOS ONE. Congratulations! Your manuscript is now being handed over to our production team.

Kind regards,

on behalf of

Dr. Gurpal S. Toor

Academic Editor

PLOS ONE